# Predictors of youth unemployment duration and impact evaluation of job creation program in East Gojjam Zone

**Nigusie Gashaye Shita**[1]*, **Metadel Azeze Mekonnen**[1], **Yeshiwas Ewinetu Tegegne**[2], **Misganaw Mekonnen Nigussie**[1], **Awoke Fetahi Woudneh**[1]

**1** Department of Statistics, Debre Markos University, Debre Markos, Ethiopia, **2** Department of Economics, Debre Markos University, Debre Markos, Ethiopia

* nigusie_gashaye@dmu.edu.et or nigusie27@gmail.com

## Abstract

Youth unemployment in the East Gojjam Zone is a critical issue. This study focuses on identifying the factors that influence unemployment duration and evaluating the impact of job creation programs on the well-being of youth in this region. We employed Cox regression to analyze the determinants of unemployment duration and used propensity score matching to assess the welfare effects of job creation initiatives. Our multistage cluster sampling revealed a youth unemployment rate of 33.3% (95% CI: 27.3–39.3). Over half of the unemployed youth transitioned to employment within four years, with 25% securing jobs within two years. Participation in job creation programs led to an average earnings increase of 1,069.716 birr, though retention in these programs was low at 49%. The findings reveal a connection between prolonged unemployment, skill mismatches, financial constraints, limited work experience, weak social networks, low income, and a preference for public-sector employment. To effectively address these challenges, interventions must focus on improving job accessibility, aligning vocational training with labor market needs, promoting financial inclusion, and enhancing social support systems.

## Introduction

Unemployment remains a critical global issue, with far-reaching effects on economic stability, social cohesion, and individual well-being [1,2]. It is the state of actively looking for work while unemployed, which includes people who have been laid off or willingly quit their jobs. Traditionally, unemployment is measured through jobless rates and duration [3,4]. Among its various forms, youth unemployment presents a particularly significant challenge. Young people often face numerous barriers when transitioning from education to employment, especially in developing regions such as sub-Saharan Africa, where the youth unemployment rate has reached a troubling 21.9% [5].

In Ethiopia, youth unemployment is a pressing concern, with urban areas experiencing a rate of 19.4% and the Amhara region at 20.2% [3,6]. East Gojjam exemplifies this crisis, where youth unemployment rose to 19.6% in 2023, a 1.1% increase from the previous year [7]. This increase can be attributed to multiple factors, including political

**Data availability statement:** All relevant data are included within the manuscript. Due to ethical considerations, including participant confidentiality and the presence of identifying information, the complete raw dataset cannot be shared publicly. Researchers seeking access to anonymized or aggregated datasets for further analysis may contact an institutional representative. Data requests will be reviewed to ensure compliance with ethical guidelines prior to sharing. For further inquiries or to request access, please contact Yihalem Abebe, Vice Dean of the College of Natural and Computational Sciences, at yihalem2000@gmail.com.

**Funding:** This research was funded by Debre Markos University, College of Natural and Computational Sciences. The funding body had no involvement in the study's design, data collection, analysis, interpretation, or the writing of the manuscript.

**Competing interests:** The authors declare that they have no competing interests.

**Abbreviations:** CIA, Conditional Independence Assumption; CSS, Central Statistical Services; ILO, International Labor Organization; MSE, Micro and Small Enterprise; PSM, Propensity score matching; ROC, Receiver Operating Characteristic.

instability, economic stagnation, a limited industrial base, an influx of migrants, and structural mismatches in the labor market [8–10]. Despite significant investments in job development programs, unemployment remains persistent, reflecting broader trends observed in other developing regions [3,11]. The sustained high unemployment not only undermines the aspirations of young people but also impedes broader regional development [12–14].

A substantial body of research has identified various factors contributing to youth unemployment. These factors include demographic characteristics such as age, gender, marital status, education level, socio-economic challenges like financial constraints, household income, limited access to job market information, and lack of credit access. Additionally, skill mismatches, insufficient training, and limited opportunities in the workplace exacerbate the problem. Institutional issues—such as corruption, unfair competition, and ineffective job search behaviors—further worsen youth unemployment. Social networks and regional disparities also significantly influence employment outcomes [15–19]. However, significant gaps remain in understanding the duration of youth unemployment and its broader implications for welfare. Most studies treat unemployment as a binary state—employed or unemployed—without examining the critical dimension of unemployment duration or the welfare outcomes of job creation initiatives [8,20–23]. Additionally, many fail to adequately address endogeneity and selection bias, raising concerns about the robustness of their findings.

Job creation programs in Ethiopia, such as the Productive Safety Net Program (PSNP) and the Urban Productive Safety Net and Jobs Project (UPSNJP), aim to address youth unemployment through skills training, job placement services, and startup capital [24–26]. However, their effectiveness has been inconsistent; varying by gender, location, and alignment with local labor market needs [24,27–29]. While well-designed vocational training and placement services can yield positive outcomes [30,31], poor implementation and misalignment with local demands often undermine their impact [32,33]. These issues highlight the need for regular evaluations and adjustments to make these programs more effective for the communities they aim to help. Despite significant investments, comprehensive assessments of these programs effectiveness in East Gojjam, particularly regarding youth unemployment duration, remain scarce.

This study seeks to fill these gaps by investigating the determinants of youth unemployment duration and assessing the impact of job creation programs in East Gojjam. Specifically, it addresses the following research questions: (1) what are the main factors influencing the duration of youth unemployment in East Gojjam? (2) How do job creation initiatives affect the duration and outcomes of youth unemployment in the region, including earnings? By employing Cox regression to identify the factors influencing unemployment duration and propensity score matching to assess the impact of job creation programs, this study aims to provide valuable insights that can guide policymakers in addressing youth unemployment in Ethiopia and similar contexts.

## Methodology

### Study design and setting

This study used a cross-sectional design conducted in the East Gojjam Zone of Amhara, Ethiopia, between September 27 and November 21, 2021. The East Gojjam Zone, situated in the Amhara Regional State, is bordered by the Oromia Region to the south, West Gojjam to the west, South Gondar to the north, and South Wollo to the east. The Abay River is a natural border, enclosing the zone on its northern, eastern, and southern sides.

## Study population and sampling methods

The study focused on unemployed youth aged 15 to 29 who registered with the East Gojjam Zone Micro and Small Enterprise (MSE) office between July 2018 and September 2021. To ensure a representative sample, we used a multistage cluster sampling method. We selected six woredas—Debre Markos, Debre Elias, Sinan, Debre Werke, Bichena, and Awabel—and within each woreda, we first applied probability proportional to size (PPS) sampling, followed by simple random sampling to select participants. We calculated the sample size at 242 based on an estimated unemployment rate of 18.5% [34], a margin of error of 4.9%, and a 95% confidence level. We excluded individuals who chose not to participate or did not provide the required information.

## Data collection methods and data quality control

The study collected primary data from selected respondents using structured questionnaires. Trained data collectors conducted in-person interviews, with telephone interviews as an alternative for unavailable respondents. Key informants also contributed valuable insights to enhance the data.

Data collectors received two days of standardized training on a pretested questionnaire and were supervised throughout the data collection process to minimize errors and bias.

To ensure data quality, we implemented several procedures. First, we conducted outlier detection to identify and address any unusual responses. We also employed effective strategies for managing missing values and performed consistency checks to ensure the responses were coherent. Additionally, we utilized data validation measures, such as cross-referencing responses, to enhance the overall reliability of the collected data.

## Study variables and operational definitions

Dependent variables included the duration of unemployment (in months) until transitioning to employment (event, censored) and earnings. The treatment variable was participation in the employment generation program (yes or no). Independent variables encompassed demographic factors (sex, age), education level, job preferences, work experience, skills, market needs, social networks, access to credit, health status, advisory services, and income.

Key terms in the study are defined as follows: Censored refers to unemployed youth who remained jobless until the data collection period in December 2021. The event is the transition of unemployed youth to employment by that date. The Employment Creation Program aims to help youth find jobs through registration in micro- and small-business offices. Long-term unemployment refers to unemployment lasting a year or more. Program non-participants (control group) are youth who remain unemployed and do not participate in the MSE employment generation program. Program participants (treatment group) are unemployed youth enrolled in the program. Short-term unemployment describes youth who are unemployed for less than a year.

## Ethics approval and consent to participate

The study received ethical approval from the Research Ethics Committee of the College of Natural and Computational Sciences at Debre Markos University, with protocol number NCS/4069/18/12, granted on September 21, 2021. We ensured that all methods were conducted according to relevant guidelines and regulations. Written informed consent was obtained from each participant before the interview. Participants were fully informed about the study's purpose, procedures, potential risks, and benefits, and they had the right to withdraw from the study at any stage without facing any restrictions.

## Data analysis

Descriptive statistics summarized the prevalence of unemployment and its associated social-demographic factors. Kaplan-Meier survival functions and log-rank tests compared the survival experiences of different groups of unemployed youth. Cox proportional hazards regression identified factors influencing unemployment duration, with assumptions verified using statistical tests (S1 Table).

Both bivariate analyses (p ≤ 0.2) and multivariate analyses (p ≤ 0.1) were performed for variable selection in the Cox regression and binary logistic models to guarantee the inclusion of pertinent variables [35–37]. The performance of these models was evaluated using likelihood ratio tests, as well as the Akaike Information Criterion (AIC) and Bayesian Information Criterion (BIC) to optimize the final models [38,39]. A p-value of less than 5% was considered significant for identifying associations between the dependent and independent variables.

Propensity score matching (PSM), which creates comparable treatment and control groups by calculating propensity scores through logistic regression, was used to evaluate the effect of the employment creation program on wages [40]. The validity of causal inferences was ensured by adhering to the assumptions of common support and conditional independence, with model performance further assessed using the Hosmer-Lemeshow test and ROC curve analysis.

## Results

### Characteristics of the study participants

The study involved 240 respondents, with a gender distribution of 68% men and 32% women. The average age was 25.14 years (±2.7), ranging from 19 to 29 years. A significant majority (63.3%) were university graduates. Access to business advisory services was limited; 68.3% reported lacking such support. Public employment was preferred by 51.2% of the youth. Among those participating in job creation programs (60%), only 12.5% had access to credit, and 38.5% achieved permanent employment. Of the 40% who did not participate, the reasons included program incompatibility (20.8%) and governance concerns (37.5%). Only 49% expressed interest in continuing with the programs (Table 1).

### Prevalence and duration of youth unemployment

The employment rate was 33.3% (95% confidence interval: 27.3–39.3). A substantial proportion, 72.5%, experienced long-term unemployment (Table 1). The median duration of unemployment was 19.2 months (±interquartile range (12.9)), ranging from 1 to 96 months. By 24 months, only 25% of young people had found employment, increasing to 48% by 48 months. The mean duration of unemployment was 18.5 months for completed spells and 25.3 months for incomplete spells. Median durations were 12 and 24 months, respectively.

### Factors associated with unemployment duration

Cox regression analysis identified several significant factors affecting unemployment duration, including skill mismatches, regional disparities, family socioeconomic status, access to credit, work experience, social networks, job preferences, and access to business advisory services (Table 2).

Higher levels of education were associated with a longer duration of unemployment (Fig 1).

In contrast, individuals with skills aligned to labor market demands experienced significantly shorter unemployment spells than those with skills mismatched to the market. Youth from low-income families faced unemployment durations that were, on average, 6.8 months longer than those from middle-class backgrounds.

**Table 1. Frequency distribution of youth unemployment in the East Gojjam Zone.**

| Variables | Category | Frequency (%) |
|---|---|---|
| Work status | Job seeker | 160 (66.7) |
| | Employed | 80 (33.3) |
| Degree of youth unemployment | Long-term unemployment | 174 (72.5) |
| | Short-term unemployment | 66 (27.5) |
| Participation in job creation programs | Yes | 144 (60) |
| | No | 96 (40) |
| Job creation program status | Temporary | 59 (61.5) |
| | Permanent | 37 (38.5) |
| Continuation status | Continuing participation | 47 (49.0) |
| | Not-continuing participation | 27 (28.1) |
| | I don't decide | 22 (22.9) |
| Reason for non-participation in job creation programs | Lack of information | 15 (15.6) |
| | Program non-compatibility | 20 (20.8) |
| | Good governance problem | 36 (37.5) |
| | Others | 25 (26.0) |
| Access to business advisory services | No | 164 (68.3) |
| | Yes | 76 (71.7) |
| Job preference | Public employment | 123 (51.2) |
| | NGO | 26 (10.8) |
| | Ownership | 78 (32.5) |
| | Partnership | 13 (5.4) |
| Access to credit | No | 84 (87.5) |
| | Yes | 12 (12.5) |
| Education status | Degree graduate | 152 (63.3) |
| | Diploma graduate | 69 (28.7) |
| | Others | 19 (7.9) |

Lack of access to credit notably extended unemployment durations, with individuals without credit access experiencing a delay of 17.9 months in securing employment. Similarly, individuals without prior work experience faced an average disadvantage of 5.0 months. Regional disparities were also observed, with youth from Debre Elias, Sinan, Debre Werke, and Bichena experiencing longer periods of unemployment compared to those in Debre Markos.

Each additional social network connection reduced unemployment duration by an average of 0.27 months. Access to business advisory services led to a 0.1-month reduction in unemployment time. Youth seeking partnerships or ownership opportunities returned to work 0.2 months faster than those aiming for public sector positions.

## Impact of the employment creation program

To assess the impact of the employment creation program on participants income, we used a logistic regression model to estimate propensity scores for matching individuals based on factors influencing program participation. Key predictors included demographic, educational, occupational, and geographic variables such as sex, age, education level, access to business consulting, prior program participation, mother's occupation, field of study, and location (S2 Table). We also included earnings-related factors, such as unemployment duration and job-related information (S3 Table), to enhance model accuracy. The final model emphasized the

**Table 2. Analysis of factors associated with unemployment duration based on the cox proportional hazards model in East Gojjam Zone.**

| Variables | Category | Hazard ratio | Standard error | p-value |
|---|---|---|---|---|
| Skill and market needs | Mismatch | 3.885 | 2.527 | 0.015 |
| | Match (ref) | | | |
| Woredas | Debre Elias | 5.39 | 3.717 | 0.015 |
| | Sinan | 6.076 | 4.097 | 0.007 |
| | Deber Werke | 4.112 | 2.666 | 0.029 |
| | Bichena | 3.983 | 2.632 | 0.037 |
| | Awabel | 2.22 | 1.564 | 0.258 |
| | Debre Markos (ref) | | | |
| Family prosperity | Poor | 6.758 | 6.758 | 0.043 |
| | Medium (ref) | | | |
| Access to credit | No | 17.866 | 17.639 | 0.004 |
| | Yes (ref) | | | |
| Work experience | No work experience | 4.954 | 2.058 | <0.001 |
| | Has work experience(ref) | | | |
| Number of social networks | Discrete variable | 0.273 | 0.149 | <0.001 |
| Job preferences | Partnership or ownership | 0.194 | 0.116 | 0.006 |
| | Others | 0.485 | 0.433 | 0.418 |
| | Public employment (ref) | | | |
| Access to business advisory services | Yes | 0.073 | 0.056 | 0.001 |
| | No (ref) | | | |

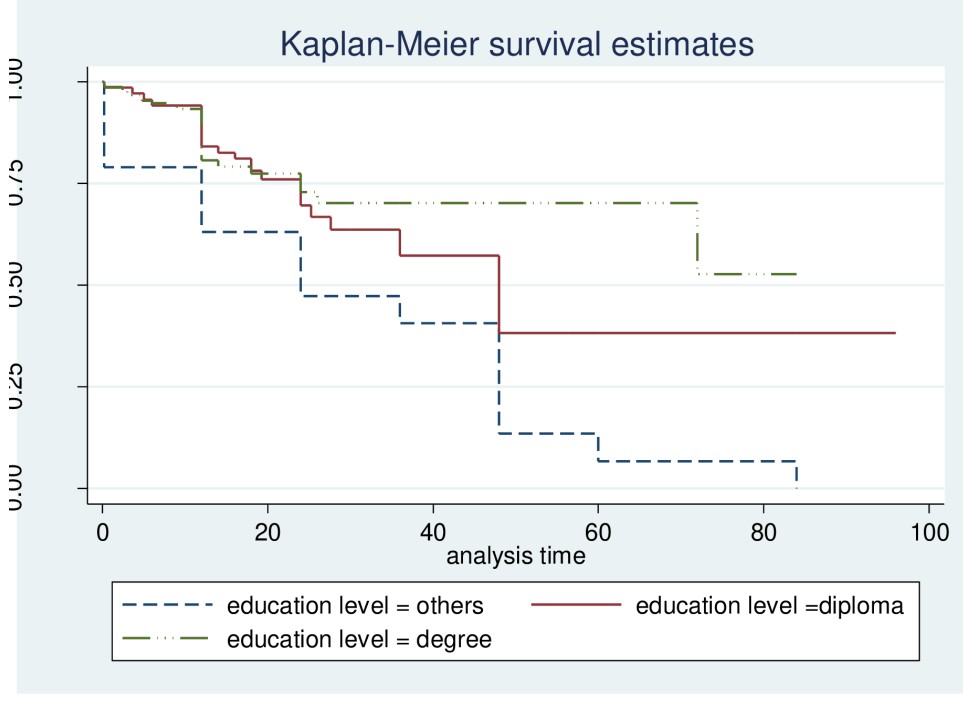

**Fig 1. Kaplan-Meier survival curve.** This Kaplan-Meier curve shows the proportion of individuals remaining unemployed over time, categorized by education level. The x-axis represents time (months), and the y-axis shows the proportion still unemployed. A steeper curve indicates faster employment rates.

importance of education level, field of study, location, gender, mother's occupation, and age in determining program participation (Table 3).

The model's robustness was confirmed through Hosmer-Lemeshow tests and ROC curve analysis (Figs 2, 3, S4 Table).

## Propensity score matching

To address potential selection bias, we implemented propensity score matching (PSM) to create comparable treatment and control groups. Nearest neighbor matching with a caliper width of 0.15 was applied, resulting in an expanded sample size from 240 to 480 individuals, evenly distributed with 240 units in each group, thereby achieving a well-balanced sample.

**Table 3. Results of the propensity score model on the participation of job creation programs in East Gojjam Zone, Northwest Ethiopia.**

| Variables | Category | Coef. | S. E. | p-value | AOR |
|---|---|---|---|---|---|
| Sex | Female | −2.239 | 0.523 | <0.001 | 0.107 |
| | Male(ref) | | | | |
| Age | Continuous | 0.362 | 0.123 | 0.003 | 1.436 |
| Education level | Certificate or below | 6.117 | 2.113 | 0.004 | 453.662 |
| | Diploma | −0.679 | 0.522 | 0.193 | 0.507 |
| | Degree (ref) | | | | |
| Field of the study | Agriculture | 5.396 | 2.361 | 0.022 | 220.462 |
| | Business economics or social science | 5.242 | 2.148 | 0.015 | 188.969 |
| | Engineering | 6.166 | 2.163 | 0.004 | 476.415 |
| | others(ref) | | | | |
| Woreda | Debre Markos | 2.911 | 0.879 | 0.001 | 18.378 |
| | Debre Elias | −4.733 | 1.277 | <0.001 | 0.009 |
| | Sinan | −4.109 | 1.167 | <0.001 | 0.016 |
| | Diver Wereke | 0.693 | 0.756 | 0.359 | 2.000 |
| | Bichena | 1.797 | 0.768 | 0.019 | 6.029 |
| | Awabel (ref) | | | | |
| Duration of unemployment | Continuous | 0.001 | 0.015 | 0.938 | 1.001 |
| Business consultant services | No | −2.787 | 0.845 | 0.001 | 0.062 |
| | Yes (ref) | | | | |
| Job preference | Public employment | 0.060 | 0.466 | 0.897 | 1.062 |
| | Non-governmental organization | 1.370 | 0.750 | 0.068 | 3.935 |
| | Partnership or ownership(ref) | | | | |
| Experience of participation in job creation programs | No | −1.320 | 0.491 | 0.007 | 0.267 |
| | Yes (ref) | | | | |
| Father's job | Government employee | 0.694 | 0.942 | 0.461 | 2.002 |
| | Run their own business | 0.037 | 0.997 | 0.970 | 1.038 |
| | Others (ref) | | | | |
| Mother's job | Government employee | 0.497 | 1.526 | 0.745 | 1.643 |
| | Run their own business | 4.733 | 1.731 | 0.006 | 113.646 |
| | Others (ref) | | | | |
| Adequate job information | No | 0.085 | 0.592 | 0.885 | 1.089 |
| | Yes (ref) | | | | |
| constant | constant | −11.274 | 3.964 | 0.004 | 0.000 |

Coef, regression coefficient; S.E, standard error; p-value, probability value; AOR, adjusted odd ratio; ref, reference category

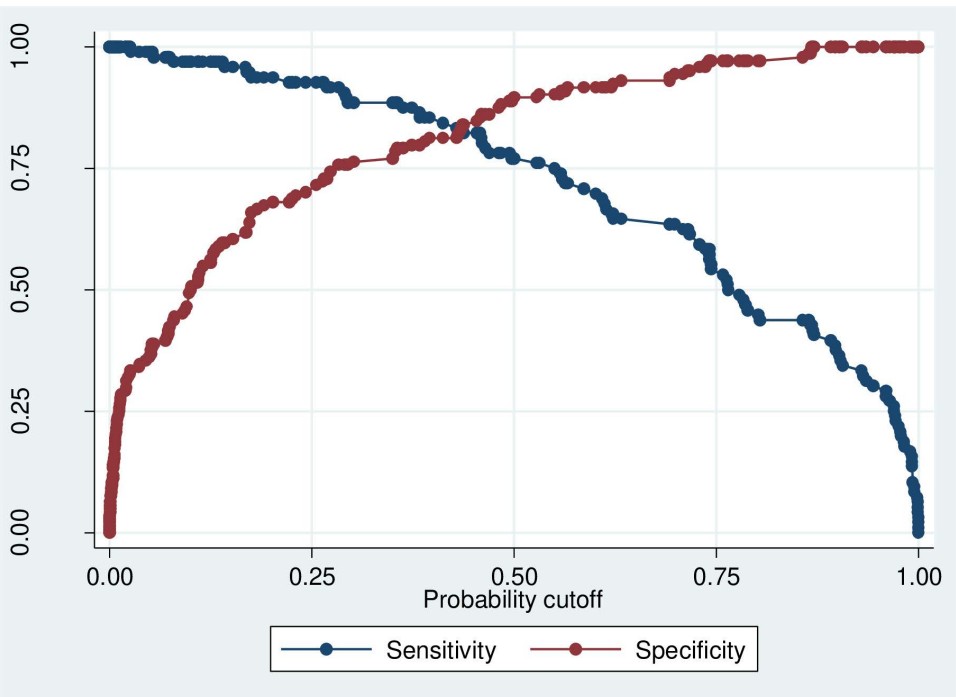

**Fig 2. Sensitivity and specificity analysis.** This figure illustrates the sensitivity and specificity of the model across different probability thresholds. The x-axis represents the threshold, and the y-axis shows the corresponding sensitivity (true positive rate) and specificity (true negative rate). A good model achieves a balance between both metrics.

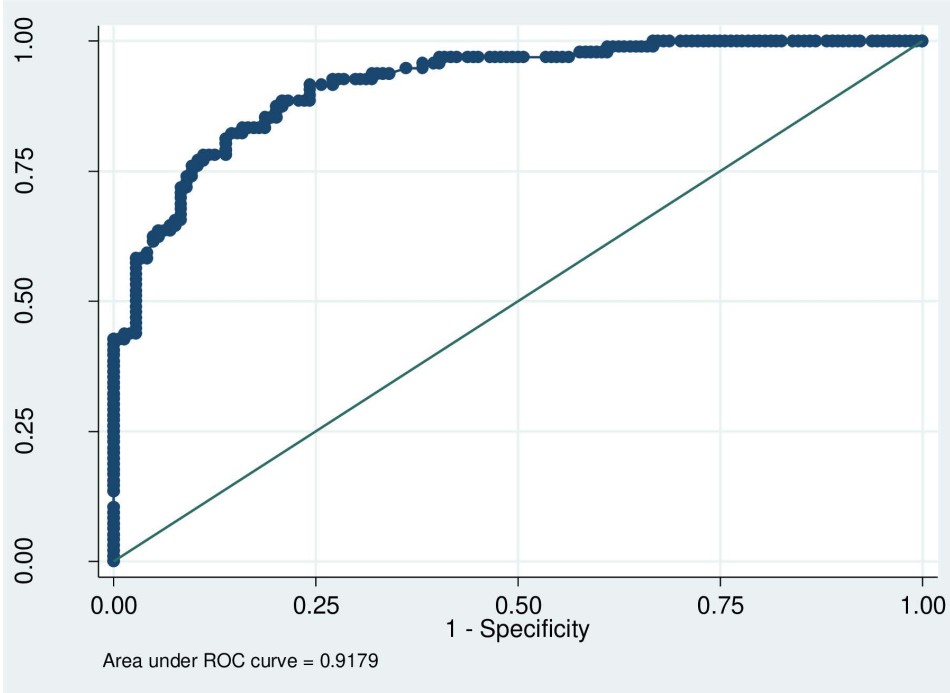

**Fig 3. Receiver operating characteristic (ROC) curve.** This ROC curve evaluates the model's diagnostic performance. The x-axis represents 1-Specificity (false positive rate), and the y-axis represents Sensitivity (true positive rate). An AUC of 0.9179 indicates excellent model performance, with a higher AUC reflecting better predictive ability.

Post-matching analysis revealed significant improvements in covariate balance, with standardized differences approaching zero and variance ratios nearing one (S5 Table). Balance plots (Fig 4) visually confirmed the reduction in covariate imbalances, and the distributions of propensity scores exhibited substantial overlap between the treatment and control groups, indicating successful matching.

Furthermore, common support was evident in the propensity score distribution (Fig 5), ensuring comparability between units in both groups.

### Evaluation of treatment effects

The analysis demonstrates a statistically significant average treatment effect (ATE) of 1,069.7 birr, indicating that individuals in the treatment group earned, on average, 1,069.7 birr more per month than those in the control group. Additionally, the average treatment effect on the treated (ATET) was 1,285.1 birr, highlighting that program participants earned an additional 1,285.1 birr per month (Table 4).

These findings provide compelling evidence of a beneficial relationship between program participation and increased earnings, affirming the programs effectiveness in enhancing economic outcomes for participants.

### Public perception of job creation programs

A survey conducted in East Gojjam, based on feedback from key informants, revealed that a significant majority of respondents (63%) viewed job creation programs as effective in fostering entrepreneurial motivation. However, opinions regarding various aspects of these

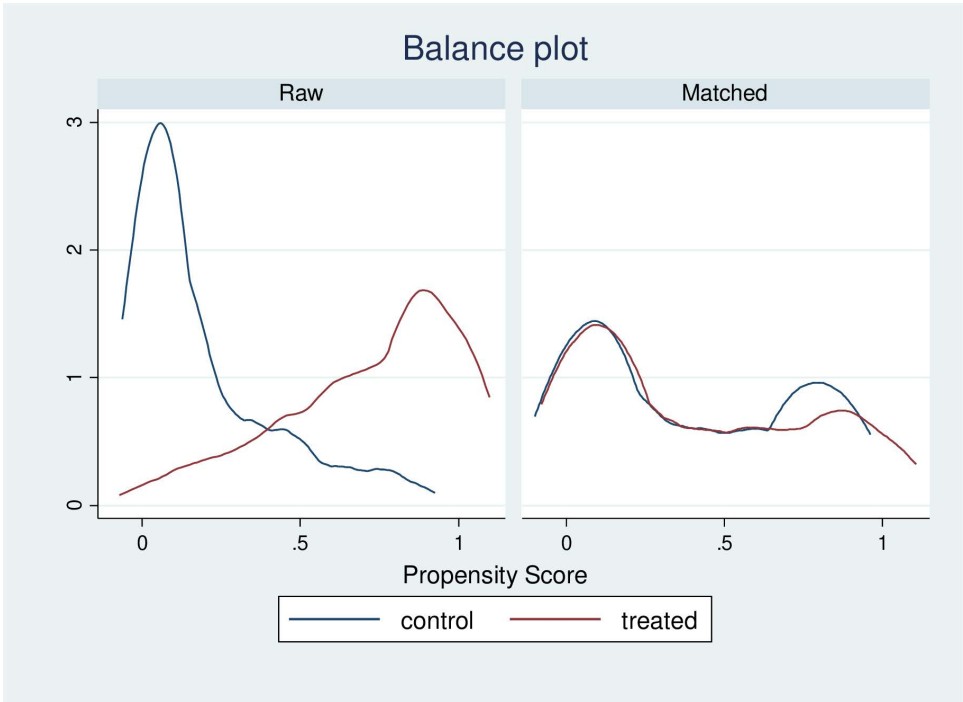

**Fig 4. Balance plot of job creation program participation.** This balance plot compares the distribution of propensity scores between the treatment and control groups. The left panel shows raw scores before matching, and the right panel shows scores after matching. Improved balance after matching enhances group comparability.

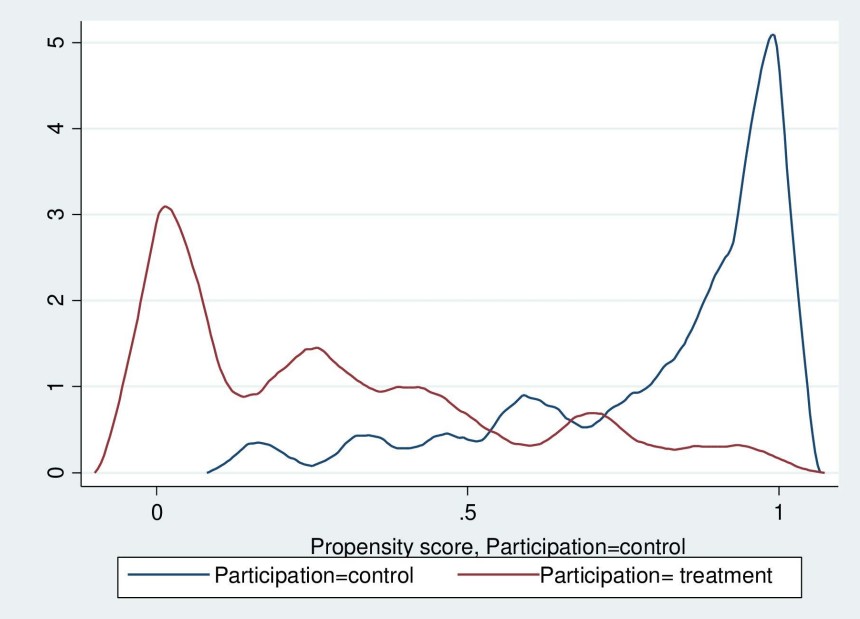

**Fig 5. Overlap plot of job creation program participation.** This overlap plot visualizes the distribution of propensity scores for the treatment and control groups. The x-axis represents the scores, and the y-axis shows density. The blue line represents the control group, and the red line represents the treatment group. Greater overlap indicates better comparability and supports the common support assumption.

programs were mixed. Many respondents acknowledged the opportunities for participation and the alignment of the programs with unemployment needs, but satisfaction levels varied. Key informants also highlighted areas for improvement, such as job access, governance, program goals, social impact, and beneficiary outcomes (Table 5).

## Discussion

Youth unemployment continues to be a significant challenge in East Gojjam, Ethiopia, with 66.7% of young people unemployed and 72.5% facing long-term unemployment, as observed in our study. While job creation programs have been implemented, they face challenges such as low participant retention, a problem also highlighted in previous research [24,41,42]. Our study found that participants in these programs experienced increased earnings and more positive perceptions of entrepreneurial motivation, aligning with findings from other Ethiopian studies [26,43]. However, challenges such as limited access to credit and insufficient job placements persist, underscoring the need for program improvements to address these ongoing issues.

Youth in our study experienced an average and median unemployment duration of 19.2 months, shorter than the 24 + months reported in previous studies [44], attributed to economic instability and limited job opportunities. This difference may indicate improvements in local employment programs and greater community and youth involvement in job development initiatives [45,46].

Youth with higher education face significantly longer unemployment durations than those with minimal education—a trend consistent with global findings [47–49] but differing from the Iranian context [50]. This disparity arises from stringent job selection criteria for highly educated individuals and insufficient job search support [47]. Despite high application rates to the public sector, job opportunities remain scarce. Additionally, the slow growth of the

**Table 4. Impact of participation on earnings in job creation programs, East Gojjam Zone, Northwest Ethiopia.**

| Effect | Earning | Coef. | Standard error | p-value |
|--------|---------|-------|----------------|---------|
| ATE | Treated vs. non-treated | 1069.716 | 208.8502 | <0.001 |
| ATET | Treated vs. non-treated | 1285.063 | 507.1094 | 0.011 |

ATE, average treatment effect; ATET, average treatment effect on the treated; Coef, Coefficient; p-value, probability value

**Table 5. Descriptive statistics for the opinions of the respondents and key formants.**

| Variable | Very low | Low | Neutral | High | Very high |
|----------|----------|-----|---------|------|-----------|
| Motivational effect (entrepreneurship) | 1 (3.7%) | 3 (11.1%) | 6 (22.2%) | 16 (59.3%) | 1 (3.7%) |
| Consumption effect | 1 (3.7%) | 12 (44.4%) | 9 (33.3%) | 5 (18.5%) | 0 (0.0%) |
| Additional support | 7 (25.9%) | 9 (33.3%) | 8 (29.6%) | 3 (11.1%) | 0 (0.0%) |
| Participation | 3 (11.1%) | 4 (14.8%) | 8 (29.6%) | 10 (37.0%) | 2 (7.4%) |
| Improving access to jobs | 3 (11.1%) | 14 (51.9%) | 4 (14.8%) | 5 (18.5%) | 1 (3.7%) |
| Compatibility with needs | 4 (14.8%) | 11 (40.7%) | 9 (33.3%) | 2 (7.4%) | 1 (3.7%) |
| Good governance effect | 3 (11.1%) | 5 (18.5%) | 6 (22.2%) | 12 (44.4%) | 1 (3.7%) |
| Social aspects strengthening | 1 (3.7%) | 5 (18.5%) | 10 (37.0%) | 10 (37.0%) | 1 (3.7%) |
| Beneficiary earnings improvement | 0 (0.0%) | 9 (33.3%) | 7 (25.9%) | 11 (40.7%) | 0 (0.0%) |
| Economic aspects strengthening | 1 (3.7%) | 8 (29.6%) | 9 (33.3%) | 9 (33.3%) | 0 (0.0%) |
| Objective achievement | 2 (7.4%) | 9 (33.3%) | 12 (44.4%) | 3 (11.1%) | 1 (3.7%) |

private sector, combined with the increasing number of graduates, exacerbates unemployment challenges for individuals with advanced degrees [51,52].

Youth with skills misaligned with labor market demands experienced significantly extended unemployment compared to peers with in-demand skills. This finding aligns with previous research [8,53–55], emphasizing that a skills mismatch significantly hinders job placement and limits access to suitable employment opportunities [52]. Conversely, possessing in-demand skills facilitates quicker job searches and improves employ-ability [56].

Youth from low-income families experienced significantly longer unemployment durations than their middle-class peers, a finding consistent with previous studies [57,58]. This finding contrasts with the U.S. context, where financial necessity often drives low-income youth to enter the labor market earlier [59]. The discrepancy may stem from the financial resources and support available to middle-income families, which can facilitate entrepreneurship and job acquisition. In contrast, the limited economic opportunities for low-income youth may restrict their access to suitable employment, prolonging their periods of unemployment [60].

Youth without access to credit faced prolonged unemployment compared to their peers with financial means. This finding aligns with previous research [61–63] and underscores the critical role of economic resources in reducing unemployment. Access to credit enables investment in entrepreneurial ventures by covering essential resources and startup costs. In contrast, the absence of financial support creates significant barriers for young entrepreneurs, prolonging their unemployment [64].

Youth without work experience faced longer unemployment durations than their peers with experience, a finding consistent with prior research [65,66]. Work experience fosters self-awareness, maturity, independence, and confidence—qualities that enhance employ-ability. Individuals with experience are often better prepared to meet market demands and adapt to job requirements, enabling them to secure employment more quickly [67].

An increase in social networks among young people correlates with shorter unemployment durations, consistent with previous studies [16,61,68,69]. Social connections are instrumental in improving job search outcomes by providing access to opportunities, career advice, and referrals, thereby enhancing employment prospects [70,71]. Conversely, limited networks can hinder job searches, leading to prolonged unemployment [72].

Youth preferring partnership or ownership experienced shorter unemployment durations than those seeking public sector jobs, aligning with previous research [73,74]. This entrepreneurial inclination can lead to faster employment transitions. However, educated youth may delay employment to secure desired formal sector positions. Conversely, those inclined toward entrepreneurship are more likely to pursue self-employment or business ventures, resulting in shorter unemployment periods [75]. These findings emphasize the importance of aligning career preferences with available opportunities to achieve quicker employment outcomes [76].

Youth who received business advisory services found jobs more quickly than those without such support, consistent with earlier research [77,78]. Business advisory services are crucial in reducing unemployment duration by helping young people navigate the job market. They provide access to work environments, skill development opportunities, and personalized counseling, all of which contribute to more efficient job searches and faster transitions into employment [79].

The study's limitation arises from the assumption that all unemployed youth are registered with small- and micro-enterprise offices in the East Gojjam zone, which may lead to an underestimation of the actual number of unemployed youth.

## Conclusions

The study in East Gojjam, northwestern Ethiopia, highlights a critical youth unemployment crisis, marked by a troubling 66.7% unemployment rate and 72.5% of young people facing long-term unemployment. While job creation initiatives have led to increased earnings, significant challenges—such as skill mismatches, regional disparities, and limited access to essential resources—continue to obstruct progress.

To effectively address these issues, a comprehensive strategy is essential, incorporating several key components. This strategy should include the development of targeted skill training programs that align with current labor market demands, ensuring that young people acquire the necessary skills for available jobs. Additionally, enhancing financial support for young entrepreneurs and job seekers is vital for promoting economic participation and driving innovation.

Strengthening governance and policy frameworks will help create a supportive environment for employment initiatives, fostering accountability and consistency in program implementation. Tailoring interventions to meet the specific needs of local communities will enhance their effectiveness, while increasing public awareness of available opportunities is crucial for motivating youth engagement in job programs and entrepreneurship. Finally, promoting gender equality will ensure equitable access to opportunities for all individuals, maximizing overall impact and fostering inclusive growth.

## Supporting information

**S1 Table.** Results of Concordance Measures, Model Fit, and Schoenfeld Residuals Test
(DOCX)

**S2 Table.** Results of multiple logistics regression on the participation of job creation programs in East Gojjam Zone, North West Ethiopia
(DOCX)

**S3 Table.** Results of multiple linear regression analysis on the predictors of earnings
(DOCX)

**S4 Table.** Results of Hosmer-Lemeshow Test
(DOCX)

**S5 Table.** Covariate Balance Assessment
(DOCX)

## Acknowledgement

We extend our sincere gratitude to the Micro and Small Enterprise (MSE) office staff of Debre Markos, Debre Elias, Sinan, Debre Wereke, Bichena, and Awabel Woredas for their invaluable support.

## Author contributions

**Conceptualization:** Nigusie Gashaye Shita, Metadel Azeze Mekonnen, Yeshiwas Ewinetu Tegegne, Misganaw Mekonnen Nigussie, Awoke Fetahi Woudneh.

**Data curation:** Nigusie Gashaye Shita, Metadel Azeze Mekonnen, Yeshiwas Ewinetu Tegegne, Misganaw Mekonnen Nigussie, Awoke Fetahi Woudneh.

**Formal analysis:** Nigusie Gashaye Shita.

**Funding acquisition:** Nigusie Gashaye Shita.

**Investigation:** Nigusie Gashaye Shita, Metadel Azeze Mekonnen, Yeshiwas Ewinetu Tegegne, Misganaw Mekonnen Nigussie, Awoke Fetahi Woudneh.

**Methodology:** Nigusie Gashaye Shita, Metadel Azeze Mekonnen, Yeshiwas Ewinetu Tegegne, Misganaw Mekonnen Nigussie.

**Project administration:** Nigusie Gashaye Shita.

**Resources:** Nigusie Gashaye Shita.

**Software:** Nigusie Gashaye Shita.

**Supervision:** Nigusie Gashaye Shita.

**Validation:** Nigusie Gashaye Shita, Metadel Azeze Mekonnen, Yeshiwas Ewinetu Tegegne, Misganaw Mekonnen Nigussie.

**Visualization:** Nigusie Gashaye Shita.

**Writing – original draft:** Nigusie Gashaye Shita.

**Writing – review & editing:** Nigusie Gashaye Shita, Metadel Azeze Mekonnen, Yeshiwas Ewinetu Tegegne, Misganaw Mekonnen Nigussie, Awoke Fetahi Woudneh.

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
