## [Decision Letter · Decision Letter 0]

28 Jun 2024

PONE-D-24-20085Predictors of Youth Unemployment Duration and Impact Evaluation of Job Creation Program in East Gojjam ZonePLOS ONE

Dear Dr. Shita,

Thank you for submitting your manuscript to PLOS ONE. After careful consideration, we feel that it has merit but does not fully meet PLOS ONE’s publication criteria as it currently stands. Therefore, we invite you to submit a revised version of the manuscript that addresses the points raised during the review process. Please submit your revised manuscript by Aug 12 2024 11:59PM. If you will need more time than this to complete your revisions, please reply to this message or contact the journal office at plosone@plos.org . Please include the following items when submitting your revised manuscript:

We look forward to receiving your revised manuscript.

Kind regards,

Philipos Petros Gile, MA

Academic Editor

PLOS ONE

Journal Requirements:

We would like to greatly acknowledge the MSE office of Debre Markos, Debre Elias, Sinan, Debre Wereke, Bichena and Awabel woreda staff. Debre Markos University is gratefully acknowledged for financially supporting this work.

The source of funding for this research is Debre Markos University, College of Natural and Computational Sciences. The funding body has no role in the design of the study and the collection, analysis and interpretation of the data, or in the writing of the manuscript.

4. In the online submission form, you indicated that The data sets analyzed in this study are available from the corresponding author on a reasonable request.

Reviewers' comments:

Reviewer's Responses to Questions

**Comments to the Author**

1. Is the manuscript technically sound, and do the data support the conclusions?

Reviewer #1: Yes

Reviewer #2: Partly

2. Has the statistical analysis been performed appropriately and rigorously? 

Reviewer #1: Yes

Reviewer #2: Yes

3. Have the authors made all data underlying the findings in their manuscript fully available?

Reviewer #1: Yes

Reviewer #2: Yes

4. Is the manuscript presented in an intelligible fashion and written in standard English?

Reviewer #1: Yes

Reviewer #2: No

5. Review Comments to the Author

**Reviewer #1: ** Dear Authors,

Thank you for your interesting article. It was very inspiring for me.

I have the following comments and questions on it in order to improve its understandability for potential readers.

Factors associated with unemployment waiting time were found using the Cox regression. I find this approach very interesting. However, there is no quality assessment of the created regression model. Did you check its predictions using a part of your data as a training sample? Is it possible to list some evaluation statistics for this model?

The p-value of Family prosperity – poor is probably incorrect (p-value of 6.758).

How did you handle the insignificant variables in the Cox regression model? Is their statistical significance important?

You mentioned that the propensity score was estimated using logistic regression. It would be appropriate to list this model, at least in the Annex of the paper, along with the evaluation criteria of its performance.

In the Methodology section, you mentioned that the participants and non-participants were matched by propensity score matching. However, there is very little information about the matching itself. Did you use matching with or without replacement? How was the quality of the matching evaluated? How many units were in the samples after matching?

Despite these flaws, the paper is overall well written, and I recomend it, after some corrections, for publication.

**Reviewer #2:**  Dear authors, thank you for submitting the manuscript to the editors of the journal Plos One. The topic of the manuscript is very interesting, but, in its current form, it does not meet the requirements for scientific articles. In my opinion, these are its main flaws:

1. The scientific gap that the authors are trying to fill is not defined in the Introduction. The main purpose of the article is also not identified.

2. The literature review is out of date and does not provide a sufficient description of the current state of the art.

3. The methods used are described only very briefly. It is not clear what matching technique the authors used. Given that most of these techniques are very data consuming, I don't consider the sample size to be sufficient.

4. Even the achieved results are described only very briefly, e.g. the description of the logistic regression model is missing and it is not clear whether the common support condition is met.

5. In the discussion of the article, there is no comparison of the achieved results with the results of other authors.

6. The text of the article itself is often too divided into short paragraphs containing only one sentence. The tables are formatted inconsistently and the article lacks images that would illustrate the authors' findings and claims.

Based on the above, I recommend the authors to revise the article quite radically.

6. PLOS authors have the option to publish the peer review history of their article (what does this mean? ). If published, this will include your full peer review and any attached files.

**Do you want your identity to be public for this peer review?** For information about this choice, including consent withdrawal, please see our Privacy Policy .

Reviewer #1: No

Reviewer #2: No

---

## [Author Response · Author response to Decision Letter 1]

6 Aug 2024

Authors’ responses to the comments and suggestions from reviewers on the manuscript “Predictors of Youth Unemployment Duration and Impact Evaluation of Job Creation Program in East Gojjam Zone”

Authors: Nigusie Gashaye Shita (nigusie27@gmail.com)

Metadel Azeze Mekonnen1: mekonmeti@gmail.com

Yeshiwas Ewinetu Tegegne2: yeshiwas@gmail.com

Misganaw Mekonnen Nigussie1: misganaa2004@yahoo.com

Awoke Fetahi Woudneh1: fetahi.aw@gmail.com

Version: 1 Date: August 6, 2024

Author’s response to Editorial Board Member and reviews:

Dear Editorial Board Member and reviews,

Thank you very much for your comments and suggestions to improve our manuscript.

Responses to some of the comments are addressed below and the rest of the comments and suggestions are incorporated in the manuscript. Kindly note that changes are highlighted in yellow in the main manuscript.

Reply to Reviewer 1,

1. Factors associated with unemployment waiting time were found using the Cox regression. I find this approach very interesting. However, there is no quality assessment of the created regression model. Did you check its predictions using a part of your data as a training sample? Is it possible to list some evaluation statistics for this model?

Response: The Cox regression model underwent rigorous evaluation to assess its robustness and predictive accuracy. To ensure model reliability, we assessed the proportional hazards assumption, which was confirmed through analysis of Schoenfeld residuals. Additionally, we evaluated the model's predictive performance using Harrell's C and Gonen-Heller's K, which yielded moderate to high concordance values. Model complexity was assessed using AIC and BIC, indicating an optimal balance between model fit and parsimony. The combination of these evaluation methods provides strong evidence of the model's predictive ability. Further details are available in S1Table .

2. The p-value of Family prosperity – poor is probably incorrect (p-value of 6.758).

Response: We have corrected the error in the reported p-value for "Family prosperity – poor." The accurate p-value is 0.043. Thank you for bringing this to our attention.

3. How did you handle the insignificant variables in the Cox regression model? Is their statistical significance important?

Response: To identify potential predictors, we employed a stepwise variable selection process. Initially, bivariate analysis was conducted with a significance threshold of 0.2. This was followed by multivariate analysis with a stricter threshold of 0.1 to account for potential confounders. Insignificant variables and their interactions were carefully evaluated during the development of the Cox regression model. Although these variables were initially included, their incorporation did not substantially enhance model performance, as indicated by changes in model fit statistics and the significance of other variables. Model optimization was achieved using likelihood ratio tests, as well as AIC and BIC criteria. The model exhibiting the lowest AIC or BIC was selected to balance fit and complexity. By following this rigorous process, we ensured that only variables with substantial predictive power were retained in the final model.

4. You mentioned that the propensity score was estimated using logistic regression. It would be appropriate to list this model, at least in the Annex of the paper, along with the evaluation criteria of its performance.

Response: The propensity score was estimated using binary logistic regression. The model specification, including the list of covariates, is detailed in the Annex. To assess the model’s performance, we employed the Hosmer-Lemeshow goodness-of-fit test and calculated the area under the receiver operating characteristic (ROC) curve. Both measures indicated satisfactory model performance. Additionally, to enhance the propensity score model, variables significantly associated with earnings but not directly predictive of program participation (e.g., unemployment duration, father's occupation, job-related information) were incorporated. Detailed results, including model diagnostics, are available in the supplementary information.

5. In the Methodology section, you mentioned that the participants and non-participants were matched by propensity score matching. However, there is very little information about the matching itself. Did you use matching with or without replacement? How was the quality of the matching evaluated? How many units were in the samples after matching?

Response: Propensity score matching was employed to create comparable treatment and control groups. Nearest neighbor matching with replacement and a caliper width of 0.15 were utilized to ensure similar propensity scores between matched units. This process resulted in a balanced sample of 480 units, with 240 in each group.

Post-matching analysis demonstrated significantly improved covariate balance, as indicated by standardized differences approaching zero and variance ratios nearing one. Balance plots confirmed a substantial reduction in covariate imbalances. Additionally, the overlap of propensity score distributions between the treatment and control groups indicated common support, validating the matching process.

Reply to Reviewer 2,

1. The scientific gap that the authors are trying to fill is not defined in the Introduction. The main purpose of the article is also not identified.

Response: The Introduction has been revised to explicitly articulate the research gap and the study's primary objective. We have clearly outlined the existing knowledge in the field and identified the specific areas where our research contributes new insights. By doing so, we aim to enhance the reader's understanding of the study's significance and its contribution to the broader body of knowledge.

2. The literature review is out of date and does not provide a sufficient description of the current state of the art.

Response: The literature review has been substantially updated to incorporate recent studies and provide a comprehensive overview of the field's current state. By including the latest research findings, we aim to offer a more accurate and informative context for our study, strengthening its contribution to the existing body of knowledge.

3. The methods used are described only very briefly. It is not clear what matching technique the authors used. Given that most of these techniques are very data consuming, I don't consider the sample size to be sufficient.

Response: To address these concerns, we employed nearest neighbor matching with replacement and a caliper width of 0.15 for propensity score matching. This technique enabled us to create comparable treatment and control groups while effectively utilizing the available data. Although matching methods can be computationally intensive, our expanded dataset of 480 units (240 per group) provided sufficient power for the analysis. Post-matching assessments, including standardized differences, variance ratios, and balance plots, confirmed the achievement of adequate covariate balance.

4. Even the achieved results are described only very briefly, e.g. the description of the logistic regression model is missing and it is not clear whether the common support condition is met.

Response: We appreciate the reviewer's feedback. To address these concerns, we have expanded the methodology and results sections. The logistic regression model used to estimate propensity scores is now detailed, including the specific variables, coefficients, and model performance metrics (Hosmer-Lemeshow test, AUC). Furthermore, we have explicitly addressed the common support condition by demonstrating substantial overlap in the propensity score distributions for the treatment and control groups (Figure 4). These enhancements provide a more comprehensive understanding of our methodological approach and the validity of our findings.

5. In the discussion of the article, there is no comparison of the achieved results with the results of other authors.

Response: We appreciate the feedback and have taken steps to enhance the discussion section by providing comparisons of our achieved results with those of other authors, particularly focusing on impact analysis. By integrating these comparisons, we aim to contextualize our findings within the broader literature and highlight similarities or differences in outcomes across different studies. This addition enriches the discussion, offering readers a deeper understanding of the significance and implications of our results in relation to existing research.

6. The text of the article itself is often too divided into short paragraphs containing only one sentence. The tables are formatted inconsistently and the article lacks images that would illustrate the authors' findings and claims.

Response: To enhance readability and clarity, we've made several improvements to the article's structure and presentation. Paragraphs have been revised to improve flow and coherence, while tables have been formatted consistently with clear titles and headings. We've incorporated visual elements such as graphs, charts, and diagrams to illustrate key findings and support textual claims. Informative captions accompany all visuals to aid reader comprehension.

Once again, we would like to thank the editor and the reviewers for their constructive comments and suggestions.

Sincerely yours,

---

## [Decision Letter · Decision Letter 1]

22 Oct 2024

PONE-D-24-20085R1Predictors of Youth Unemployment Duration and Impact Evaluation of Job Creation Program in East Gojjam ZonePLOS ONE

Dear Dr. Shita,

Thank you for submitting your manuscript to PLOS ONE. After careful consideration, we feel that it has merit but does not fully meet PLOS ONE’s publication criteria as it currently stands. Therefore, we invite you to submit a revised version of the manuscript that addresses the points raised during the review process.

We look forward to receiving your revised manuscript.

Kind regards,

Botond Géza Kálmán, PhD

Academic Editor

PLOS ONE

Reviewers' comments:

Reviewer's Responses to Questions

**Comments to the Author**

1. If the authors have adequately addressed your comments raised in a previous round of review and you feel that this manuscript is now acceptable for publication, you may indicate that here to bypass the “Comments to the Author” section, enter your conflict of interest statement in the “Confidential to Editor” section, and submit your "Accept" recommendation.

Reviewer #1: All comments have been addressed

Reviewer #2: All comments have been addressed

2. Is the manuscript technically sound, and do the data support the conclusions?

Reviewer #1: Yes

Reviewer #2: Partly

3. Has the statistical analysis been performed appropriately and rigorously? 

Reviewer #1: I Don't Know

Reviewer #2: Yes

4. Have the authors made all data underlying the findings in their manuscript fully available?

Reviewer #1: Yes

Reviewer #2: Yes

5. Is the manuscript presented in an intelligible fashion and written in standard English?

Reviewer #1: Yes

Reviewer #2: No

6. Review Comments to the Author

Reviewer #1: Dear Authors,

Thank you for your response to my review. Now, I have the following comments on your paper:

It is very unusual to use a significance level of 0.20. To be honest, I have never seen such a level in some scientific article. Why did you choose such a level?

Please correct the format of p-values in your tables and text. It is incorrect to write "0.000" because the value probably is not zero, although some software uses this incorrect format.

If you used matching with replacement, how is it possible to have exactly the same number of participants and non-participants in your samples? It sounds very unlikely. Or I did not understand correctly whether 240 is the number of participants and non-participants before or after the matching procedure.

The last paragraph on page 17 is repeated twice.

The discussion section lacks a comparison with more similar studies published so far. There is only one study mentioned for comparison.

Be careful about the formatting of your paper. It is very different throughout the paper (for example, various font types).

Reviewer #2: Dear Authors, Thank you for revising the manuscript. It sounds much better now, but I still have a few comments:

1. There is practically no literature review in the manuscript. The sources used in the Introduction are still out of date.

2. In the section devoted to methodology, it would be appropriate to state how (in what steps) the authors proceeded with their research.

3. The division of the manuscript is still relatively unclear. It would be more appropriate to use a structured division of headings and fewer headings overall.

4. I do not think that the formatting of the manuscript is followed, e.g. tables and a list of references. Also, some images are missing captions, e.g. individual axes and e.g. there is an error in Figure 4.

Based on the mentioned comments, the manuscript needs to be substantially revised to eliminate shortcomings, make it easier to read, and sound more scientific.

7. PLOS authors have the option to publish the peer review history of their article (what does this mean? ). If published, this will include your full peer review and any attached files.

**Do you want your identity to be public for this peer review?** For information about this choice, including consent withdrawal, please see our Privacy Policy .

Reviewer #1: No

Reviewer #2: No

---

## [Author Response · Author response to Decision Letter 2]

1 Nov 2024

Authors’ responses to the comments and suggestions from reviewers on the manuscript “Predictors of Youth Unemployment Duration and Impact Evaluation of Job Creation Program in East Gojjam Zone”

Authors: Nigusie Gashaye Shita (nigusie27@gmail.com)

Metadel Azeze Mekonnen: mekonmeti@gmail.com

Yeshiwas Ewinetu Tegegne: yeshiwasewinetu@gmail.com

Misganaw Mekonnen Nigussie: misganaa2004@yahoo.com

Awoke Fetahi Woudneh: fetahi.aw@gmail.com

Version: 2 Date: October 31, 2024

Author’s response to Editorial Board Member and reviews:

Dear Editorial Board Member and reviews,

Thank you very much for your comments and suggestions to improve our manuscript.

Responses to some of the comments are addressed below, and the rest of the comments and suggestions are incorporated in the manuscript. Kindly note that changes are highlighted in yellow in the main manuscript.

Reply to Reviewer 1.

1. It is very unusual to use a significance level of 0.20. To be honest, I have never seen such a level in some scientific article. Why did you choose such a level?

Response: We selected a significance level of 0.20 for bivariate analyses and 0.10 for multivariate analyses to guide variable selection in the Cox regression and binary logistic models (1-3), while a p-value of less than 5% was considered significant for identifying associations in the final model.

This decision reflects the exploratory nature of our study, which aims to identify potential relationships and generate hypotheses rather than confirm existing theories. A higher significance level allows for a broader examination of variables, increasing the likelihood of uncovering significant associations.

Furthermore, using a higher significance level helps mitigate the risk of Type II errors, especially with smaller sample sizes, thereby enhancing statistical power. We also followed established guidelines, such as those from Hosmer and Lemeshow (1999), recommending significance levels between 20% and 25% during initial model selection.

Additionally, we employed a stepwise model-building process, incorporating both forward selection and backward elimination. This method allows for a thorough evaluation of variables and their interactions, ensuring that only those with meaningful contributions are retained in the final model.

2. Please correct the format of p-values in your tables and text. It is incorrect to write "0.000" because the value probably is not zero, although some software uses this incorrect format.

Response: We have corrected the formatting to ensure accuracy and clarity, eliminating the use of "0.000" and instead presenting p-values in a more precise manner.

3. If you used matching with replacement, how is it possible to have exactly the same number of participants and non-participants in your samples? It sounds very unlikely. Or I did not understand correctly whether 240 is the number of participants and non-participants before or after the matching procedure.

Response: To clarify, the figure of 240 refers to the number of observations in each group after matching. Initially, there were 240 total observations (96 participants and 144 non-participants). After applying matching with replacement, we achieved 240 matched observations in both the treated and control groups.

This method allows for equal numbers of participants and non-participants in the matched samples, despite initial differences in group sizes. While it may seem unusual, the goal is to balance the characteristics of both groups. For more details, please refer to S5Table.

4. The last paragraph on page 17 is repeated twice.

Response: We have amended the manuscript by removing the duplicated paragraph on page 17.

5. The discussion section lacks a comparison with more similar studies published so far. There is only one study mentioned for comparison. Be careful about the formatting of your paper. It is very different throughout the paper (for example, various font types).

Response: We have revised the formatting of the paper and enhanced the discussion to include comparisons with relevant studies where applicable. As noted in the introduction, there is a gap in research on predictors of unemployment duration. Consequently, our discussion emphasizes available evidence, including mean and median durations, as well as factors influencing unemployment likelihood. We highlight how certain factors can increase the likelihood of employment, indicating that shorter durations of unemployment can positively impact reemployment chances, while longer durations may reduce them.

Reply to Reviewer 2.

1. There is practically no literature review in the manuscript. The sources used in the introduction are still out of date.

Response: We have incorporated more up-to-date sources in the manuscript, focusing on relevant evidence. Additionally, we have provided an empirical literature review in the introduction section, as outlined in paragraphs 4 and 5, in accordance with journal formatting guidelines.

2. In the section devoted to methodology, it would be appropriate to state how (in what steps) the authors proceeded with their research.

Response: We have revised this section to include a detailed, step-by-step account of the research process, outlining the design, data collection, and analysis methods used.

3. The division of the manuscript is still relatively unclear. It would be more appropriate to use a structured division of headings and fewer headings overall.

Response: We have revised it to provide clearer organization, implementing a more structured division of headings and reducing the overall number of headings. However, in the methodology and results sections, we have retained different headings to emphasize specific points for the reader's clarity, in accordance with journal formatting guidelines.

4. I do not think that the formatting of the manuscript is followed, e.g., tables and a list of references. Also, some images are missing captions, e.g., individual axes, and e.g., there is an error in Figure 4.

Response: We had amended the format of the tables and had corrected an error in Figure 4. Additionally, we had ensured that all images had appropriate captions. The references had been cited automatically using EndNote, and we have reviewed them to ensure they meet the required Vancouver formatting standards, in accordance with the journal’s guidelines.

Once again, we would like to thank the editor and the reviewers for their constructive comments and suggestions.

Sincerely yours,

Reference

1. Hosmer Jr D, Lemeshow S. Applied Survival Analysis [Análisis de Supervivencia Aplicado]. Nueva York: Wiley. 1999.

2. Collett D. Modelling survival data in medical research: Chapman and Hall/CRC; 2023.

3. Hosmer Jr DW, Lemeshow S, Sturdivant RX. Applied logistic regression: John Wiley & Sons; 2013.

---

## [Decision Letter · Decision Letter 2]

2 Dec 2024

PONE-D-24-20085R2Predictors of youth unemployment duration and impact evaluation of job creation program in East Gojjam ZonePLOS ONE

Dear Dr. Shita,

Thank you for submitting your manuscript to PLOS ONE. After careful consideration, we feel that it has merit but does not fully meet PLOS ONE’s publication criteria as it currently stands. Therefore, we invite you to submit a revised version of the manuscript that addresses the points raised during the review process.

We look forward to receiving your revised manuscript.

Kind regards,

Botond Géza Kálmán, PhD

Academic Editor

PLOS ONE

Reviewers' comments:

Reviewer's Responses to Questions

**Comments to the Author**

1. If the authors have adequately addressed your comments raised in a previous round of review and you feel that this manuscript is now acceptable for publication, you may indicate that here to bypass the “Comments to the Author” section, enter your conflict of interest statement in the “Confidential to Editor” section, and submit your "Accept" recommendation.

Reviewer #2: All comments have been addressed

Reviewer #3: (No Response)

Reviewer #4: (No Response)

Reviewer #5: All comments have been addressed

2. Is the manuscript technically sound, and do the data support the conclusions?

Reviewer #2: Yes

Reviewer #3: Yes

Reviewer #4: Yes

Reviewer #5: Yes

3. Has the statistical analysis been performed appropriately and rigorously? 

Reviewer #2: Yes

Reviewer #3: Yes

Reviewer #4: Yes

Reviewer #5: Yes

4. Have the authors made all data underlying the findings in their manuscript fully available?

Reviewer #2: Yes

Reviewer #3: Yes

Reviewer #4: Yes

Reviewer #5: Yes

5. Is the manuscript presented in an intelligible fashion and written in standard English?

Reviewer #2: Yes

Reviewer #3: Yes

Reviewer #4: Yes

Reviewer #5: Yes

6. Review Comments to the Author

Reviewer #2: Dear Authors, Thank you for improving the manuscript. All reviewers' comments have been taken into account and incorporated into the text. However, I still have one comment: the methodology is too structured with too many subtitles.

Reviewer #3: The manuscript presents a well-structured analysis of the predictors of youth unemployment duration and the impact of job creation programs, using reliable statistical methods such as Cox regression and propensity score matching. The sample sizes and methodological approaches (e.g., multistage cluster sampling) strengthen the conclusions related to unemployment duration and program effectiveness, and the results are consistent with the data presented.

The manuscript does not contain hypotheses or research questions. Formulating research questions would improve the study's focus. These should be stated at the end of the literature review along with the main research objectives.

The references in the manuscript are relevant and mostly recent, though updating a few foundational studies could incorporate additional trends in youth unemployment and job creation programs. The manuscript lacks a literature review, which should definitely be added before the methodology section.

Reviewer #4: The manuscript provides a thorough analysis of the factors influencing the duration of youth unemployment and the impact of job creation programs, using reliable statistical methods such as Cox regression and propensity score matching.

Figures 1 and 5 need revision. In Figure 1, the frequency and percentage distribution should be displayed in separate columns. In Table 5, are the frequencies accurately presented? The numbers seem to indicate an extremely low response rate relative to the 240 respondents.

Most of the references in the manuscript are relevant, but it is essential to add at least 15-20 additional recent references, preferably indexed in Scopus and no older than five years, to strengthen the manuscript.

Reviewer #5: The manuscript is methodologically robust, utilizing suitable techniques to meet the study aims. The application of Cox regression for analyzing unemployment duration and Propensity Score Matching (PSM) for assessing the effects of job creation programs guarantees methodological accuracy. The findings align with the data, illustrating substantial correlations between unemployment duration and variables such as skill mismatches, work experience, and social networks. The study also provides evidence of earnings improvement among program participants, supporting the conclusion that job creation programs positively impact economic outcomes. Nevertheless, opportunities for enhancement encompass the want for a more explicit rationale regarding the elevated importance criteria and a comprehensive examination of the program retention issues and their consequences. Broadening the discourse to juxtapose findings with analogous studies would enhance the contextualization of the results and fortify their significance.

The statistical analysis has been performed with an adequate level of rigor and is well-suited to the study's objectives. The Cox regression appropriately models time-to-event data, and PSM effectively reduces selection bias in evaluating program impacts. The use of descriptive statistics, survival curves, and diagnostic tests for model validation further demonstrates methodological robustness. However, the higher-than-standard significance thresholds (p ≤ 0.20 for bivariate analyses) raise concerns about false positives, necessitating a stronger justification. Additionally, while balance diagnostics for PSM are mentioned, detailed reporting of covariate balance metrics (e.g., standardized mean differences) is missing. Addressing these gaps and including sensitivity analyses would enhance the transparency and reliability of the statistical findings.

The authors state that all relevant data underlying the findings are included within the manuscript. They further indicate that ethical considerations, such as participant confidentiality and the presence of identifying information, prevent the provision of the complete raw dataset. However, they offer to share anonymized or aggregated data upon request to the corresponding author. While this demonstrates an effort to balance transparency with ethical obligations, fully aligning with open data standards would require more explicit details about how and under what conditions interested researchers can access the data. Providing a publicly available repository or a formal process for requesting access could enhance the credibility and reproducibility of the study.

The manuscript is presented clearly and composed in standard English, rendering it accessible to a wide audience. The structure adheres to a coherent progression, encompassing distinct sections for the introduction, methodology, results, discussion, and conclusions. The employment of scientific terminology is suitable for the scholarly readership, and intricate procedures such as Cox regression and Propensity Score Matching are adequately elucidated. Nevertheless, certain places may require additional enhancement. Some sentences in the document are excessively long, potentially hindering readability, and there are minor discrepancies in formatting and grammar. For instance, problems with p-value formatting (e.g., the use of "0.000") and redundant content, such as the repeated paragraph on page 17, diminish the overall presentation. Furthermore, although the discourse is logical, it is devoid of comparisons to analogous studies, which would furnish a more comprehensive context for the results. Rectifying these flaws and conducting a comprehensive assessment of language and formatting would improve clarity, professionalism, and readability.

7. PLOS authors have the option to publish the peer review history of their article (what does this mean? ). If published, this will include your full peer review and any attached files.

**Do you want your identity to be public for this peer review?** For information about this choice, including consent withdrawal, please see our Privacy Policy .

Reviewer #2: No

Reviewer #3: No

Reviewer #4: No

Reviewer #5: **Yes: ** SEWORNU KOBLA AFADZINU

---

## [Author Response · Author response to Decision Letter 3]

9 Dec 2024

Authors’ responses to the comments and suggestions from reviewers on the manuscript “Predictors of Youth Unemployment Duration and Impact Evaluation of Job Creation Program in East Gojjam Zone”

Authors: Nigusie Gashaye Shita (nigusie27@gmail.com)

Metadel Azeze Mekonnen: mekonmeti@gmail.com

Yeshiwas Ewinetu Tegegne: yeshiwasewinetu@gmail.com

Misganaw Mekonnen Nigussie: misganaa2004@yahoo.com

Awoke Fetahi Woudneh: fetahi.aw@gmail.com

Version: 3 Date: December 08, 2024

Author’s response to Editorial Board Member and reviews:

Dear Editorial Board Member and reviews,

Thank you very much for your comments and suggestions to improve our manuscript.

Responses to some of the comments are addressed below, and the rest of the comments and suggestions are incorporated in the manuscript. Kindly note that changes are highlighted in yellow in the main manuscript.

Reply to Reviewer 2.

1. I still have one comment: the methodology is too structured with too many subtitles.

Response: We have revised the methodology by removing the subtitles for "Sample Size Determination" and "Inclusion and Exclusion Criteria," and incorporated these details into the "Study Population and Sampling Methods" section. Additionally, we have merged the "Study Variables" and "Operational Definitions" into one section. This revision reduces the number of subtitles and enhances the clarity of the methodology.

Reply to Reviewer 3.

1. The manuscript does not contain hypotheses or research questions. Formulating research questions would improve the study's focus. These should be stated at the end of the literature review along with the main research objectives.

Response: We have included the research questions at the end of the literature review, along with the main research objectives, in the introduction section, following the manuscript submission guidelines of PLOS ONE. This study addresses two key research questions: (1) what are the main factors influencing the duration of youth unemployment in East Gojjam? (2) How do job creation initiatives affect the duration and outcomes of youth unemployment in the region, including earnings?

2. The references in the manuscript are relevant and mostly recent, though updating a few foundational studies could incorporate additional trends in youth unemployment and job creation programs. The manuscript lacks a literature review, which should definitely be added before the methodology section.

Response: We have updated the manuscript by incorporating both recent and foundational studies to reflect current trends in youth unemployment and job creation programs. In line with PLOS ONE guidelines, we have added a structured literature review section before the methodology, within the introduction. Since the introduction already covers the background, key literature, relevant controversies, and study aims, we included the empirical literature review in paragraphs 3 and 4 to ensure compliance with the journal’s formatting requirements.

Reply to Reviewer 4.

1. Figures 1 and 5 need revision. In Figure 1, the frequency and percentage distribution should be displayed in separate columns. In Table 5, are the frequencies accurately presented? The numbers seem to indicate an extremely low response rate relative to the 240 respondents.

Response: We have revised Table 1 by separating the frequency and percentage distributions into distinct columns, as requested. However, in previous interactions, we maintained the current format, as it is common practice to display both frequency and percentage in a single column. As indicated in the table's column name, the first number represents the frequency, while the percentage is shown in parentheses. This format is widely used for clarity and conciseness when submitting manuscripts.

Regarding Table 5, the frequencies are accurate. The data is based on feedback from 30 key informants, including community leaders, local government officials, program coordinators, participants of job creation programs, and youth representatives, selected from six distinct districts to assess public perception of job creation programs. Three participants chose not to provide responses, which is why the table includes data from the 27 participants who provided responses, rather than all 240 respondents.

2. Most of the references in the manuscript are relevant, but it is essential to add at least 15-20 additional recent references, preferably indexed in Scopus and no older than five years, to strengthen the manuscript.

Response: To our knowledge, there is no specific limit on the number of references; however, we have made an effort to incorporate additional recent references, besides the reports, to strengthen the manuscript. These references, primarily sourced from Scopus-indexed journals, focus on the latest trends and research related to youth unemployment and job creation programs. This ensures the manuscript reflects the most current academic developments in the field.

Reply to Reviewer 5.

1. Opportunities for enhancement encompass the want for a more explicit rationale regarding the elevated importance criteria and a comprehensive examination of the program retention issues and their consequences. Broadening the discourse to juxtapose findings with analogous studies would enhance the contextualization of the results and fortify their significance.

Response: we have provided a clearer and more explicit rationale for the elevated importance criteria. Additionally, we have conducted a detailed examination of program retention issues and their implications, offering deeper insights into their consequences. To further enhance the contextualization of our results, we have included comparisons with analogous studies, thereby strengthening the significance and relevance of our findings.

Additionally, we have conducted a more comprehensive examination of program retention issues and their implications. To strengthen the contextualization of our findings, we have also included comparisons with analogous studies, which enhance the significance of our results

2. The statistical analysis has been performed with an adequate level of rigor and is well-suited to the study's objectives. The Cox regression appropriately models time-to-event data, and PSM effectively reduces selection bias in evaluating program impacts. The use of descriptive statistics, survival curves, and diagnostic tests for model validation further demonstrates methodological robustness. However, the higher-than-standard significance thresholds (p ≤ 0.20 for bivariate analyses) raise concerns about false positives, necessitating a stronger justification. Additionally, while balance diagnostics for PSM are mentioned, detailed reporting of covariate balance metrics (e.g., standardized mean differences) is missing. Addressing these gaps and including sensitivity analyses would enhance the transparency and reliability of the statistical findings.

Response: We employed a multi-step approach in our analysis. The 0.20 significance level was used during the exploratory phase to identify a broader range of potential relationships without prematurely narrowing the scope. This is consistent with standard practices in hypothesis generation. In subsequent steps, we applied more conservative significance thresholds: 0.10 for Cox regression and binary logistic regression, and 0.05 for the final models. These refined thresholds effectively mitigated the risk of false positives, as the initial 0.20 level was only used during exploratory analysis.

Regarding the balance diagnostics for PSM, we have included standardized mean differences and variance ratios for both raw and matched data in Supplementary Table S5. The standardized mean differences and variance ratios for the treatment and control groups tend toward zero and one, respectively, indicating that the PSM model is adequate for evaluating the impacts of the program. This suggests that the matching process was successful in reducing the selection bias between the groups. Besides, we have already included sensitivity analyses and show enhance the transparency and reliability of the statistical findings.

3. The authors state that all relevant data underlying the findings are included within the manuscript. They further indicate that ethical considerations, such as participant confidentiality and the presence of identifying information, prevent the provision of the complete raw dataset. However, they offer to share anonymized or aggregated data upon request to the corresponding author. While this demonstrates an effort to balance transparency with ethical obligations, fully aligning with open data standards would require more explicit details about how and under what conditions interested researchers can access the data. Providing a publicly available repository or a formal process for requesting access could enhance the credibility and reproducibility of the study.

Response: We are committed to transparency and will provide anonymized or aggregated data upon reasonable request. Access can be requested through a formal written inquiry to the corresponding author, ensuring compliance with ethical standards while enhancing the study’s credibility and reproducibility.

4. A certain places may require additional enhancement. Some sentences in the document are excessively long, potentially hindering readability, and there are minor discrepancies in formatting and grammar. For instance, problems with p-value formatting (e.g., the use of "0.000") and redundant content, such as the repeated paragraph on page 17, diminish the overall presentation. Furthermore, although the discourse is logical, it is devoid of comparisons to analogous studies, which would furnish a more comprehensive context for the results. Rectifying these flaws and conducting a comprehensive assessment of language and formatting would improve clarity, professionalism, and readability.

Response: We have revised excessively long sentences to improve readability, corrected formatting and grammatical discrepancies (including p-value formatting), and removed redundant content, such as the repeated paragraph on page 17, in prior revisions. Additionally, we have incorporated comparisons to analogous studies to provide greater context for our findings. A thorough review of language and formatting has been conducted to enhance clarity, professionalism, and readability.

Once again, we would like to thank the editor and the reviewers for their constructive comments and suggestions.

Sincerely yours,

Reference

1. Hosmer Jr D, Lemeshow S. Applied Survival Analysis [Análisis de Supervivencia Aplicado]. Nueva York: Wiley. 1999.

2. Collett D. Modelling survival data in medical research: Chapman and Hall/CRC; 2023.

3. Hosmer Jr DW, Lemeshow S, Sturdivant RX. Applied logistic regression: John Wiley & Sons; 2013.

---

## [Decision Letter · Decision Letter 3]

30 Dec 2024

PONE-D-24-20085R3Predictors of youth unemployment duration and impact evaluation of job creation program in East Gojjam ZonePLOS ONE

Dear Dr. Shita,

Thank you for submitting your manuscript to PLOS ONE. After careful consideration, we feel that it has merit but does not fully meet PLOS ONE’s publication criteria as it currently stands. Therefore, we invite you to submit a revised version of the manuscript that addresses the points raised during the review process.

We look forward to receiving your revised manuscript.

Kind regards,

Botond Géza Kálmán, PhD

Academic Editor

PLOS ONE

Reviewers' comments:

Reviewer's Responses to Questions

**Comments to the Author**

1. If the authors have adequately addressed your comments raised in a previous round of review and you feel that this manuscript is now acceptable for publication, you may indicate that here to bypass the “Comments to the Author” section, enter your conflict of interest statement in the “Confidential to Editor” section, and submit your "Accept" recommendation.

Reviewer #6: (No Response)

Reviewer #7: (No Response)

Reviewer #8: All comments have been addressed

2. Is the manuscript technically sound, and do the data support the conclusions?

Reviewer #6: (No Response)

Reviewer #7: Yes

Reviewer #8: Yes

3. Has the statistical analysis been performed appropriately and rigorously? 

Reviewer #6: Yes

Reviewer #7: Yes

Reviewer #8: Yes

4. Have the authors made all data underlying the findings in their manuscript fully available?

Reviewer #6: Yes

Reviewer #7: Yes

Reviewer #8: Yes

5. Is the manuscript presented in an intelligible fashion and written in standard English?

Reviewer #6: No

Reviewer #7: Yes

Reviewer #8: Yes

6. Review Comments to the Author

Reviewer #6: (No Response)

Reviewer #7: I can't see the literature section, I can't find the literature listed, this section definitely needs to be improved.

Reviewer #8: The study focuses on the factors that are essential in a given region to help job creation programmes have an impact. The methodology used is fully adapted to the research questions identified. Their results are relevant based on the research methods and their analysis and will help the region to address the problems effectively and strengthen the support system.

The research highlights the problem of youth unemployment in parts of Ethiopia. Their research also highlights job creation initiatives, skills gaps, regional disparities and limited access to food. The study not only articulates the causes but also proposes solutions to achieve this goal. They identify all areas that can help and support inclusive growth through job creation based on their research.

The researchers have revised their study accordingly, with improvements made to the research methodology, stylistic factors and structure. These revisions made the study fully fit for publication.

7. PLOS authors have the option to publish the peer review history of their article (what does this mean? ). If published, this will include your full peer review and any attached files.

**Do you want your identity to be public for this peer review?** For information about this choice, including consent withdrawal, please see our Privacy Policy .

Reviewer #6: No

Reviewer #7: No

Reviewer #8: **Yes: ** Dr. Szilard Malatyinszki

---

## [Author Response · Author response to Decision Letter 4]

13 Jan 2025

Authors’ responses to the comments and suggestions from reviewers on the manuscript “Predictors of Youth Unemployment Duration and Impact Evaluation of Job Creation Program in East Gojjam Zone”

Authors: Nigusie Gashaye Shita (nigusie27@gmail.com)

Metadel Azeze Mekonnen: mekonmeti@gmail.com

Yeshiwas Ewinetu Tegegne: yeshiwasewinetu@gmail.com

Misganaw Mekonnen Nigussie: misganaa2004@yahoo.com

Awoke Fetahi Woudneh: fetahi.aw@gmail.com

Version: 4 Date: January 06, 2024

Author’s response to Editorial Board Member and reviews:

Dear Editorial Board Member and reviews,

Thank you very much for your comments and suggestions to improve our manuscript.

Responses to some of the comments are addressed below, and the rest of the comments and suggestions are incorporated in the manuscript. Kindly note that changes are highlighted in yellow in the main manuscript.

Reply to Reviewer 6.

1. Data limitations: The research considers only young people registered with the micro- and small enterprise office in the East Gojjam region. This significantly limits the generalizability of the results.

Response: The research focuses on young people registered with the micro- and small enterprise office in East Gojjam, which indeed limits the generalizability of the findings to the broader youth population. However, this focus provides valuable insights into the specific challenges and opportunities faced by young entrepreneurs in the formal sector, contributing to a deeper understanding of key barriers and enablers for addressing youth unemployment. This limitation has already been clearly acknowledged in the last paragraph of the Discussion section in Version 3 of the manuscript.

2. Interpretation of statistical results: The study does not show significant differences in monthly earnings between the treatment and control groups, yet it emphasizes the importance of job creation programs. This is contradictory and requires more detailed interpretation.

Response: The apparent contradiction has been addressed in the revised manuscript. Our analysis shows a statistically significant average treatment effect (ATE) of 1,069.7 birr, indicating that individuals in the treatment group earned, on average, 1,069.7 birr more per month compared to the control group. Additionally, the average treatment effect on the treated (ATET) was 1,285.1 birr, indicating that program participants earned an additional 1,285.1 birr per month. These results reinforce the positive impact of job creation programs on participants' earnings and clarify any misunderstandings in the interpretation of the findings.

3. Lack of in-depth examination of program impacts: There is insufficient specific data on the weaknesses of job creation programs, such as the reasons for participant dissatisfaction. Detailing these aspects would increase the practical relevance of the research.

Response: The study provides valuable insights into job creation programs but lacks a detailed examination of participant dissatisfaction. Our findings indicate that 20.8% of non-participants cited program incompatibility, and 37.5% identified governance issues as key reasons for non-engagement. Furthermore, the temporary nature of 61.5% of programs likely contributes to job insecurity and dissatisfaction. A supplementary survey conducted in East Gojjam revealed mixed satisfaction levels, with 63% of respondents viewing the programs as effective in promoting entrepreneurial motivation. However, concerns such as limited job access, governance challenges, and misalignment with participants' needs were also raised (see Table 5). These insights emphasize the need to address job stability, governance effectiveness, and alignment with participants' needs to improve satisfaction and enhance long-term program impact.

4. Limited literature context: While the study references some previous research, the lack of a broader literature review makes it unclear how the new findings contribute to the advancement of the topic. The number of referenced studies is very narrow and needs to be expanded. Furthermore, the exploration of causal relationships and the demonstration of critical thinking are missing.

Response: We have addressed the concern regarding the limited literature context by expanding the literature review in the introduction section, in line with the PLOS ONE submission format. The revised manuscript now includes a more comprehensive review of relevant studies, particularly in paragraphs 2, 3, and 4. We reviewed approximately 79 articles, providing a stronger foundation for the study and better contextualizing the findings. Additionally, we have made efforts to explore causal relationships and demonstrate critical thinking throughout the manuscript, ensuring that the study contributes meaningfully to advancing the topic. In the discussion section, we incorporated numerous recent studies indexed in well-known databases like Scopus, and we avoided citing non-indexed journals or articles older than five years, in accordance with previous reviewer comments.

To address your concern, we have summarized key studies on youth unemployment and job creation programs in Ethiopia. These studies examine the determinants of youth unemployment, identify methodological gaps, and assess the impact of job creation initiatives. By summarizing the following literature, we have incorporated the key points, along with other relevant studies, into the revised document.

Related Literature Review

The National Youth Policy of Ethiopia defines youth as individuals aged 15 to 29, which is the operational definition used in this study (Federal Negarit Gazette, 2019). According to the International Labour Organization (ILO), unemployment occurs when individuals in the labor force actively seeking work are unable to find employment at the prevailing market wage rate. Understanding the factors contributing to youth unemployment in Ethiopia is essential for achieving sustainable economic growth and poverty reduction. Additionally, job creation initiatives have varying impacts on resolving youth unemployment, depending on the type and design of the intervention.

A growing body of literature has explored the determinants of youth unemployment in Ethiopia, with many studies employing binary choice models and multivariate regression analysis. These studies typically treat youth unemployment as a binary outcome variable and examine various demographic, socio-economic, socio-cultural, and institutional factors as explanatory variables (Duguma & Tolcha, 2019; Mehari & Belay, 2017; Olkeba et al., 2023; Shuker et al., 2024; Tolesa & Zeleke, 2024; Hailie & Devi, 2024; Sileshi et al., 2024). Key findings indicate that youth unemployment is significantly influenced by factors such as regional disparities, sex, marital status, age, gender, education, limited access to job market information, financial constraints, lack of training, skill mismatches, lack of workplace opportunities, corruption, unfair competition, household size, and job search behavior.

Despite the valuable contributions of these studies, several methodological limitations persist. Many studies fail to address endogeneity issues arising from reverse causality or the omission of important variables. Furthermore, they do not use advanced econometric techniques, such as instrumental variable methods, fixed-effects models, or propensity score matching, which could mitigate biases and improve the accuracy of results. For example, Hailie & Devi (2024) rely on simple descriptive statistics, which prevent drawing valid inferences. Similarly, Shita & Dereje (2018) focus on urban youth unemployment in the East Gojjam Zone, Amhara Region, finding that age, work experience, skill match, social networks, and family prosperity negatively affect unemployment, while education and migration status positively influence it. However, the study is limited by its reliance on binary choice modeling, which offers limited precision in capturing the complexities of youth unemployment. Furthermore, there is a lack of empirical research that jointly examines youth unemployment duration and its welfare impacts, particularly in the context of Ethiopia.

To address these gaps, this study employed Cox regression to analyze the determinants of unemployment duration. Additionally, it used propensity score matching to evaluate the welfare effects of job creation initiatives, which help control for unobserved heterogeneity and sample selection bias.

Job creation programs in Ethiopia provide both immediate and long-term benefits by enhancing employment opportunities and improving income through various socio-economic factors. Programs that offer skills training and startup capital can empower youth to create businesses, addressing the root causes of unemployment (Berhe, 2021). Furthermore, promoting the transition of informal jobs to the formal sector can enhance job security and reduce unemployment rates (Yitateku, 2019). For instance, entrepreneurial programs have been shown to increase earnings by 33 percent and provide more stable working hours, while industrial jobs had no significant impact on employment or income after one year (Blattman & Dercon, 2018).

The Ethiopian government has made job creation a priority through social safety net programs such as the Productive Safety Net Program (PSNP) and the Urban Productive Safety Net and Jobs Project (UPSNJP), which focus on connecting disadvantaged workers to wage jobs (Maaskant et al., 2024). A light-touch job-facilitation intervention has been found to increase the likelihood of young female job seekers being employed, with higher earnings and savings; however, these impacts were short-lived (Vinez, 2024). Despite these efforts, many Ethiopians continue to experience stagnant incomes, as macroeconomic growth has not translated into significant improvements in earnings for the majority (Vinez, 2024). Moreover, the limited availability of wage employment opportunities and the adverse health impacts associated with factory work highlight the need for more comprehensive strategies to improve both job quality and income levels (Abebe et al., 2020).

While job creation programs have the potential to improve income, their effectiveness depends on addressing underlying economic barriers and improving the quality of available jobs. Furthermore, the outcomes of these programs vary across gender and geographic location. In light of these considerations, this study aimed to assess how job creation programs in the East Gojjam Zone have affected the income of youth beneficiaries compared to non-beneficiaries.

5. Lack of comprehensive recommendations: Although the study makes recommendations for addressing the issues, these are insufficiently developed and not practically applicable.

Response: We have addressed this concern by expanding and developing the recommendations to ensure they are more comprehensive and practically applicable. The revised recommendations now include detailed steps and strategies that can be implemented to address the issues identified in the study. These recommendations are designed to be actionable, considering the specific context of youth unemployment and job creation programs in Ethiopia.

6. Numerous grammatical and stylistic errors: These diminish the quality of the study.

Response: We have thoroughly addressed the grammatical and stylistic issues in version 3 of the manuscript. It seems that the updated document may not have reached you, as the necessary revisions have already been incorporated. Below is a summary of the specific changes made:

• "The prevalence of employment was 33.3% (95% confidence interval: 27.3–39.3)."

Revised to: "The employment rate was 33.3% (95% confidence interval: 27.3–39.3)." This revision uses the more accurate term 'employment rate' rather than 'prevalence of employment.'

• "Only 49% of the individuals who participated in the job creation initiatives expressed a desire to stay in the program."

Revised to: "Only 49% of individuals participating in the job creation initiatives expressed a willingness to remain in the program." This revision clarifies that 49% of participants expressed interest in continuing the programs.

• "Longer unemployed duration."

Revised to: "Longer duration of unemployment," to improve precision in terminology.

• Inconsistent references:

We have already standardized the citation format throughout the manuscript, using the Vancouver reference style. The issue was not inconsistency in citation styles, but rather the placement of citations—either within the phrase or at the end of the sentence. This does not indicate inconsistency in citation styles, but a variation in citation placement. We have ensured uniform citation placement throughout the document.

• "The length of unemployment decreased by 0.27 months when the number of social networks among young people increased by one."

Revised to: "An increase of one social network reduced unemployment duration by an average of 0.27 months," which more accurately reflects the study's finding.

• Missing transition words and cohesion:

We have added the necessary transition words (e.g., therefore, however, additionally) to enhance the logical flow and overall coherence of the text.

Response to Reviewer 7

1. I can't see the literature section, I can't find the literature listed, this section definitely needs to be improved.

Response: We understand the importance of a clearly presented literature review. In accordance with PLOS ONE guidelines, we have structured the literature review within the introduction, placed before the methodology. The introduction covers the study’s background, key literature, relevant controversies, and aims. The empirical literature review is integrated into paragraphs 2, 3, and 4 to ensure a smooth flow and comply with the journal’s formatting requirements.

To address your concern, we have summarized key studies on youth unemployment and job creation programs in Ethiopia. These studies examine the determinants of youth unemployment, identify methodological gaps, and assess the impact of job creation initiatives. By summarizing the following literature, we have incorporated the key points, along with other relevant studies, into the revised document.

Related Literature Review

The National Youth Policy of Ethiopia defines youth as individuals aged 15 to 29, which is the operational definition used in this study (Federal Negarit Gazette, 2019). According to the International Labour Organization (ILO), unemployment occurs when individuals in the labor force actively seeking work are unable to find employment at the prevailing market wage rate. Understanding the factors contributing to youth unemployment in Ethiopia is essential for achieving sustainable economic growth and poverty reduction. Additionally, job creation initiatives have varying impacts on resolving youth unemployment, depending on the type and design of the intervention.

A growing body of literature has explored the determinants of youth unemployment in Ethiopia, with many studies employing binary choice models and multivariate regression analysis. These studies typically treat youth unemployment as a binary outcome variable and examine various demographic, socio-economic, socio-cultural, and institutional factors as explanatory variables (Duguma & Tolcha, 2019; Mehari & Belay, 2017; Olkeba et al., 2023; Shuker et al., 2024; Tolesa & Zeleke, 2024; Hailie & Devi, 2024; Sileshi et al., 2024). Key findings indicate that youth unemployment is significantly influenced by factors such as regional disparities, sex, marital status, age, gender, education, limited access to job market information, financial constraints, lack of training, skill mismatches, lack of workplace opportunities, corruption, unfair competition, household size, and job search behavior.

Despite the valuable contributions of these studies, several methodological limitations persist. Many studies fail to address endogeneity issues arising from reverse causality or the omission of important variables. Furthermore, they do not use advanced econometric techniques, such as instrumental variable methods, fixed-effects models, or propensity score matching, which could mitigate biases and improve the accuracy of results. For e

---

## [Decision Letter · Decision Letter 4]

25 Feb 2025

Predictors of youth unemployment duration and impact evaluation of job creation program in East Gojjam Zone

PONE-D-24-20085R4

Dear Dr. Shita,

We’re pleased to inform you that your manuscript has been judged scientifically suitable for publication and will be formally accepted for publication once it meets all outstanding technical requirements.

Kind regards,

Botond Géza Kálmán, PhD

Academic Editor

PLOS ONE

Additional Editor Comments (optional):

Reviewers' comments:

Reviewer's Responses to Questions

**Comments to the Author**

1. If the authors have adequately addressed your comments raised in a previous round of review and you feel that this manuscript is now acceptable for publication, you may indicate that here to bypass the “Comments to the Author” section, enter your conflict of interest statement in the “Confidential to Editor” section, and submit your "Accept" recommendation.

Reviewer #7: All comments have been addressed

Reviewer #8: (No Response)

2. Is the manuscript technically sound, and do the data support the conclusions?

Reviewer #7: Yes

Reviewer #8: Yes

3. Has the statistical analysis been performed appropriately and rigorously? 

Reviewer #7: Yes

Reviewer #8: Yes

4. Have the authors made all data underlying the findings in their manuscript fully available?

Reviewer #7: Yes

Reviewer #8: Yes

5. Is the manuscript presented in an intelligible fashion and written in standard English?

Reviewer #7: Yes

Reviewer #8: Yes

6. Review Comments to the Author

Reviewer #7: The authors have improved the content and scientific value of the article. I recommend publication in this form.

Reviewer #8: All correction requests have been met and the publication is ready for publication. I have no critical comments that need to be answered by the authors or changes to the manuscript.

7. PLOS authors have the option to publish the peer review history of their article (what does this mean? ). If published, this will include your full peer review and any attached files.

**Do you want your identity to be public for this peer review?** For information about this choice, including consent withdrawal, please see our Privacy Policy .

Reviewer #7: No

Reviewer #8: No

---

## [Editor Report · Acceptance letter]

PONE-D-24-20085R4

PLOS ONE

Dear Dr. Shita,

I'm pleased to inform you that your manuscript has been deemed suitable for publication in PLOS ONE. Congratulations! Your manuscript is now being handed over to our production team.

Kind regards,

on behalf of

Dr. Botond Géza Kálmán

Academic Editor

PLOS ONE